# Genomic Comparative Analysis of Two Multi-Drug Resistance (MDR) *Acinetobacter baumannii* Clinical Strains Assigned to International Clonal Lineage II Recovered Pre- and Post-COVID-19 Pandemic

**DOI:** 10.3390/biology12030358

**Published:** 2023-02-24

**Authors:** German Matias Traglia, Fernando Pasteran, Jenny Escalante, Brent Nishimura, Marisel R. Tuttobene, Tomás Subils, Maria Rosa Nuñez, María Gabriela Rivollier, Alejandra Corso, Marcelo E. Tolmasky, Maria Soledad Ramirez

**Affiliations:** 1Departamento de Desarrollo Biotecnológico, Instituto de Higiene, Facultad de Medicina, Universidad de la Republica, Montevideo 11200, Uruguay; 2National Regional Reference Laboratory for Antimicrobial Resistance (NRL), Servicio Antimicrobianos, Instituto Nacional de Enfermedades Infecciosas, ANLIS Dr. Carlos G. Malbrán, Buenos Aires 1282, Argentina; 3Center for Applied Biotechnology Studies, Department of Biological Science, College of Natural Sciences and Mathematics, California State University Fullerton, Fullerton, CA 92831, USA; 4Instituto de Biología Molecular y Celular de Rosario (IBR, CONICET-UNR), Rosario 2000, Argentina; 5Instituto de Procesos Biotecnológicos y Químicos de Rosario (IPROBYQ, CONICET-UNR), Rosario 2000, Argentina; 6Laboratorio de Microbiología, Hospital Provincial Neuquén Dr. Castro Rendón, Neuquén 8300, Argentina; 7Laboratorio de Microbiología, Hospital Artémides Zatti, Viedma, Rio Negro 8500, Argentina

**Keywords:** *Acinetobacter baumannii* MDR, ST-2, COVID-19, *bla*
_OXA-23_, *bla*
_NDM-1_

## Abstract

**Simple Summary:**

*Acinetobacter baumannii* is a problematic bacterium that causes hard-to-treat hospital infections worldwide. Multiple cases of *A. baumannii*/SARS-CoV-2 co-infection were reported during the pandemic. This fact raised the question of whether the strains in those co-infections had or acquired unique genetic traits. This study is a comparative analysis of two strains from the same clonal group, but one was isolated before the pandemic, and the other was isolated from a patient with COVID-19. Their genomes had a high similarity, indicating that they may have derived from a unique background. However, each genome had numerous unique genes that were involved in virulence and resistance to antimicrobials. These differences could result from adaptative evolution to the human body infected with SARS-CoV-2.

**Abstract:**

Background: After the emergence of COVID-19, numerous cases of *A. baumannii*/SARS-CoV-2 co-infection were reported. Whether the co-infecting *A. baumannii* strains have distinctive characteristics remains unknown. Methods and Results: *A. baumannii* AMA_NO was isolated in 2021 from a patient with COVID-19. AMA166 was isolated from a mini-BAL used on a patient with pneumonia in 2016. Both genomes were similar, but they possessed 337 (AMA_NO) and 93 (AMA166) unique genes that were associated with biofilm formation, flagellar assembly, antibiotic resistance, secretion systems, and other functions. The antibiotic resistance genes were found within mobile genetic elements. While both strains harbored the carbapenemase-coding gene *bla*_OXA-23_, only the strain AMA_NO carried *bla*_NDM-1_. Representative functions coded for by virulence genes are the synthesis of the outer core of lipooligosaccharide (OCL5), biosynthesis and export of the capsular polysaccharide (KL2 cluster), high-efficiency iron uptake systems (acinetobactin and baumannoferrin), adherence, and quorum sensing. A comparative phylogenetic analysis including 239 additional sequence type (ST) 2 representative genomes showed high similarity to *A. baumannii* ABBL141. Since the degree of similarity that was observed between *A. baumannii* AMA_NO and AMA166 is higher than that found among other ST2 strains, we propose that they derive from a unique background based on core-genome phylogeny and comparative genome analysis. Conclusions: Acquisition or shedding of specific genes could increase the ability of *A. baumannii* to infect patients with COVID-19.

## 1. Introduction

Infections that are caused by *Acinetobacter* are associated with mortality rates as high as 60% [1,2,3,4]. Multi-drug resistance (MDR) isolates are becoming more common, and many of them include carbapenems as the antibiotics to which they are immune [5,6]. Consequently, the World Health Organization (WHO) and the Centers for Disease Control and Prevention (CDC) have designated *A. baumannii* as a high-priority pathogen for antibiotic research and development.

*A. baumannii* causes a variety of infections, such as ventilator-associated pneumonia, bloodstream, and catheter-associated urinary tract infections [7,8]. This bacterium is often part of polymicrobial infections, which are common in hospitalized patients [9,10]. Gram-negative organisms can take advantage of being a component of polymicrobial infections [11,12,13,14]. Co-infections of *Acinetobacter* with other bacterial, fungi, or viral pathogens complicate the design of successful therapies [15,16,17,18]. Our recent results showed that *A. baumannii* and *Staphylococcus aureus* can coexist at the site of infection, producing a more aggressive process [19]. *A. baumannii* can sense and undergo phenotypic changes in response to molecules that are secreted by *S. aureus* [20]. Other kinds of co-infections are those involving viruses and bacteria, which raise significant medical concerns. Co-infections that are caused by different bacteria and the SARS-CoV-2 virus were widely reported [21,22,23,24,25,26]. A study found that OXA-23-producing carbapenem-resistant sequence type (ST) 2 *A. baumannii* (ST2-CRAB) isolates were the etiologic agent of outbreaks among COVID-19 patients at the ICU in Tehran, Iran [27]. *A. baumannii*/SARS-CoV-2 co-infection has not been reported yet in South America. The present study aimed to analyze and compare two ST2 *A. baumannii* clinical strains from Argentina. One of them was isolated from a patient before the pandemic (AMA166), and the other from a COVID-19 patient (AMA_NO). Extensive genomic analysis suggested that the two strains derive from a common ancestor. The studies also identified genetic determinants that shaped the strains’ antimicrobial resistance profile. The differences in resistance profiles between both strains could result from their adaptative evolution.

## 2. Materials and Methods

### 2.1. Bacterial Isolates

*A. baumannii* AMA166 was isolated in 2016 (Argentina) from a 64-year-old patient with type 2 diabetes and arterial hypertension. The *A. baumannii* AMA_NO strain was isolated in 2021 (Argentina) from a 58-year-old patient with COVID-19. Both patients were hospitalized and received colistin as monotherapy. Both strains were cultured in LB medium and were identified using MALDI-TOF MS [28].

### 2.2. Genome Sequencing

Genomic DNA was extracted using the DNeasy Blood and Tissue kit (Qiagen Germantown, MD, USA) following the manufacturer’s instructions. Whole genome sequencing was carried out on the NextSeq 550 Illumina (MiGS sequencing service). Quality control of sequencing was performed using the FASTQ software. De novo assembly and quality assessment were done with the SPAdes and the QUAST software, respectively [29,30]. The Whole Genome Shotgun project has been deposited at GenBank with accession numbers JANKJZ000000000 and JANKKA000000000 for AMA166 and AMA_NO, respectively.

### 2.3. Comparative Genome Analysis

Genome annotation of both strains was performed using the PROKKA software [31]. The ortholog functional assignment was done using EggNOG v2.0 (default parameter) [32]. The taxonomy assignment was performed by ANI% with the seven reference genomes from *A. baumannii-calcoaceticus* complex using the JSpeciesWS software using the default parameter [33].

The tRNAscan-SE and Infernal software were used for tRNA and ncRNA prediction [34]. The Multilocus sequence typing (MLST) profile was determined using MLST scripts (https://github.com/tseemann/mlst, accessed on 9 February 2023). The antimicrobial resistance genes (ARG) were identified using the BLASTp software and the databases CARD-RGI (e-value < 10^−6^, Amino Acid Identity > 30%, Coverage > 70%) [35]. Identification of virulence factors was carried out using BLASTp and the database VFDB (Virulence Factor Database) (e-value < 10^−6^, Amino Acid Identity > 30%, Coverage > 70%) [36]. The K and OC loci were identified using the Kaptive software using the default parameters [37]. The high-affinity iron-uptake locus was identified using BLASTp (e-value < 10^−6^, Amino Acid Identity > 30%, Coverage > 70%). Nucleotide sequences of each iron-uptake system were taken from Antunes et al. [38]. Insertion sequences were determined using BLASTp and the ISFinder database (e-value< 10^−6^, Amino Acid Identity > 30%, Coverage > 70%) [39]. The prophages were predicted using the PHASTER Software using the default parameters [40]. The presence of plasmids of different groups was carried out by *rep* and *mob* homology analysis (e-value < 10^−6^) [41].

To determine the core-genome, the initial datasets of the 2938 ST2 genomes were downloaded from NCBI GenBank (January 2022). The core-genome genes were obtained using the Roary software using the default parameters (Appendix A) [42]. A total of 722 core genes were concatenated and aligned using the MAFFT software (parameter: --auto) [43]. The core-genome sequences were aligned and clustered by nucleotide identity (nucleotide identity > 98%) using the “Decipher” R package [44]. One reference genome of each cluster was selected from a total of 239 clusters (Appendix A). All the selected genomes were included in the study to perform the core-genome phylogeny analysis.

The core genome phylogeny analysis was performed applying the maximum likelihood method using the RAxML software using the default parameters [45]. The substitution genetic model was done by JModelTest2 using the default parameters [46]. SNPs were extracted using the snp-sites software using the default parameters [47]. The genes that were unique to each genome were extracted from “gene_presence_absence.csv” of Roary output.

The co-linearity analysis was performed using the progressive mauve algorithm with default parameters [48].

### 2.4. RNA Extraction and qRT-PCR Analysis

The *A. baumannii* AMA_NO strain was cultured in LB broth for 24 h at 37 °C with shaking. RNA extraction was performed using the Direct-zol RNA miniprep Kit (ZYMO research, Irvine, CA, USA). Quantification of RNA was performed using a DeNovix DS-11+ Spectrophotometer. RNA quality was assessed on a 1.5% agarose gel via gel electrophoresis. DNase treatment was performed following the manufacturer’s instructions (Thermo Fisher Scientific, Waltham, MA, USA) and quantified as previously described [49,50]. The absence of DNA was confirmed by PCR amplification of the 16S rDNA gene. Reverse transcription was carried out using the iScript Reverse Transcription Supermix for qRT-PCR (BioRad, Hercules, CA, USA) according to the manufacturer’s instructions. qRT-PCR was performed using iQ™SYBR Green Supermix (BioRad, Hercules, CA, USA) per the manufacturer’s recommendations. Specific oligonucleotides to amplify *bla*_OXA-23_ and *bla*_NDM-1_ were used. Transcriptional levels of each sample were normalized to the transcriptional level of *rpoB*. The relative quantification of gene expression was performed using the comparative threshold method 2^−ΔCt^ [51]. The reactions were performed in four technical and three biological replicates, respectively. The statistical analysis (*t*-test) was performed using GraphPad Prism (GraphPad software, San Diego, CA, USA). A *p*-value < 0.05 was considered significant.

### 2.5. Antibiotic Susceptibility Assays

The antimicrobial resistance profile was determined by disk diffusion (10 µg ampicillin/sulbactam, 30 µg amikacin, 30 µg cefepime, 10 µg ceftazidime 5 µg ciprofloxacin, 10 µg imipenem, 10 µg gentamicin, 10 µg meropenem, 15 µg tigecycline, 30 µg minocycline, or 10 µg colistin) and minimum inhibition concentration determination according to the Clinical and Laboratory Standards Institute (CLSI) recommendations [52]. The experiments were repeated at least three times for each strain. The results were interpreted with CLSI guidelines, except for colistin and tigecycline, in which cases European Committee on Antimicrobial Susceptibility Testing (EUCAST) and Food and Drug Administration (FDA) recommendations were used, respectively. The CLSI, EUCAST, and FDA publish guidelines for antimicrobial susceptibility testing (AST) that provide recommendations for testing and interpreting the susceptibility of microorganisms to antimicrobial agents. These guidelines include recommendations for standardized methods, quality control procedures, and interpretive criteria for AST [52,53].

## 3. Results and Discussion

### 3.1. Sequencing of AMA166 and AMA_NO. Genomic and Phylogenomic Comparative Analyses

Strains AMA_NO and AMA166, isolated post- and pre-COVID pandemic, were used to study changes at the genomic level to identify gene acquisition that could indicate patterns of adaptative genomic evolution [54,55]. The source of the AMA_NO strain was a patient with COVID-19. The whole genome sequences of both strains were of good quality, and depth was greater than 50X coverage (Table 1). The assembly quality was evaluated using the QUAST software and produced 79 (AMA_NO) and 38 (AMA166) contigs [30]. Both genomes had similar sizes with a difference of 254.427 bp between them. The N50 of AMA_NO and AMA166 were 125.130 and 220.435, respectively. The numbers of tRNAs identified were 61 (AMA_NO) and 60 (AMA 166) (Table 1).

Mobile genetic elements, such as insertion sequences (ISs), prophages, and plasmids, play an important role in genome evolution and the dissemination of antimicrobial resistance genes in *A. baumannii*. ISs were identified using the BLAST algorithm and the ISFinder database [39]. A total of three ISs were common in both bacteria, but strain AMA_NO possessed nine more (Appendix A). Also, the disrupted transposases of ISs (as pseudogene) were found in both genomes. There were twelve and two disrupted IS transposases that were found in AMA_NO and AMA166 genomes, respectively (Appendix A).

Analysis using the PHASTER online software found four prophages that were common to both strains (Appendix A). The PHASTER prediction classification detected one intact category prophage, one incomplete category prophage, and two questionable category prophages in both genomes. Plasmids were not found when searching for known *rep* and *mob* genes [41].

A total of 41 and 38 non-coding RNA (ncRNA) regulatory elements were identified in strains AMA_NO and AMA166, respectively (Table 1). The three ncRNA that were present only in strain AMA_NO were ALILL, group-II-D1D4-2, and Intron_gpII (Appendix A). ALILL pseudoknot is an RNA element that induces frameshifting. This element was identified through comparative analysis of a class of transposable elements belonging to the IS3 family and is conserved across bacterial species, such as *Lactobacillus lactis*, *Escherichia coli*, *Acinetobacter* species, etc. [56,57,58]. Group II introns (group-II-D1D4-2 and Intron_gpII) are a large class of self-catalytic ribozymes and mobile genetic elements that are found within the genes of all three domains of life. Remarkably, AMA_NO contains seven disrupted IS3 family transposase genes (Appendix A). Considering these results, we hypothesize that the Group II introns might have had regulatory functions of the IS3-family insertion sequences activity in ancestors of the AMA_NO strain. The presence of incomplete IS3 family sequences could be due to a reduction of the genome and a possible path towards a greater specialization to occupy one or more specific ecological niches.

The MLST profile was determined using the Pasteur and Oxford scheme. AMA_NO and AMA166 belong to the ST2 (Pasteur)/ST208 (Oxford) clone (Clonal Complex 2). With the Oxford scheme, duplication of *gdhB* was identified, corresponding to alleles 3 and 189 (paralogous genes). Considering Gaiarsa et al.’s report, the allele 3 was considered, assigned the ST208 as the MLST profile [59]. A core-genome phylogenetic analysis of ST2 (Pasteur) clone showed two main phylogenetic clusters (A and B) (Figure 1). Both strains clustered together in the phylogenetic cluster A with high similarity with the *A. baumannii* ABBL141 strain isolated in the USA (Figure 1). In the *A. baumannii* ABLL141, identified as ST208 in the scheme Oxford, the same duplication of *gdhB* alleles that were found in our strains (AMA_NO and AMA166) is present (Appendix A).

A comparison of gene content showed that 3546 genes were shared by both genomes, while 269 and 14 were unique to AMA_NO and AMA166, respectively (Appendix A). In addition, co-linearity analysis was performed to evaluate the genome structural variation between AMA_NO and AMA166. There were six local collinear blocks (LCBs), which indicate genomic regions with the highest homology, that were determined between both genomes. LCB 1 was observed inverted and translocated in AMA166, while LCB 4 and 5 were translocated in AMA166 (Appendix A).

To assess the possible biological activity or function of unique genes, functional annotation was performed using the EggNog Mapper v 2.0 software [32]. At least one category was assigned to 237 out of 269 and 10 out of 14 unique genes of AMA_NO and AMA166, respectively (Appendix A). The potential functions of the unique genes in AMA166 were transposases, hydrolases, and monoxidases. Some possible functions of the unique genes in AMA_NO were related to virulence, such as folate biosynthesis (*folKP, sul1, sul2*), pyruvate metabolism (*adh, frmA, adhP, pdhD*), biofilm formation (*impC*), and tyrosine metabolism (*adh, frmA, adhP*) (Appendix A). These results indicate that post-pandemic *A. baumannii* ST2 may have acquired virulence-associated genes, which suggests an increase in pathogenicity and resistance to treatment (Appendix A). Similar observations were found in other opportunistic pathogens, such as *Klebsiella pneumoniae* and *Aspergillus* [60,61].

### 3.2. Antibiotic Resistance, Virulence, and Its Association with Horizontal Genetic Transfer (HGT) Elements

A total of 18 and 10 ARG were identified in strains AMA_NO and AMA166, respectively (Figure 2). The intrinsic β-lactamases *bla*_OXA-66_ and *bla*_ADC-25_ were present in both genomes (Figure 2); according to their genetic environment, they do not seem to be associated with a transposable element. Both strains include a copy of Tn*2008*, a transposon that contains the *bla*_OXA-23_ gene. Identical Tn*AbaR* multidrug resistance genomic islands harboring *tetB* (tetracycline resistance), *strA,* and *strB* (streptomycin resistance) were found in strains AMA_NO and AMA166 genomes (Figure 3). This Tn*AbaR* possesses an identical resistance gene array to those in 907 out of the 6702 *A. baumannii* genomes deposited in GenBank at the time of this study. When considering only the *A. baumannii* ST2 clone genomes, the Tn*AbaR* present in strains AMA166 and AMA_NO was found in 202 out of 2938 genomes currently in GenBank (Figure 3). There were three efflux pumps, AdeABC, AdeFGH, AdeIJK, and their regulator systems that were found in both genomes. These efflux pumps are associated with fluorquinolone, carbapenem, cephalosporin, phenicol, macrolide, tetracycline, rifampicin, glycylcycline, and lincosamide resistance [62,63,64]. Also, the ade efflux pumps are associated with biofilm formation, fitness, and pathogenesis [62]. Both genomes contain the class 1 integron with the same gene cassette array, *aacC1-orfP-orfQ-aadA1*, in the variable region. This gene cassette array is present in 167 of the 2938 ST2 *A. baumannii* genomes in GenBank.

The ARGs *bla*_NDM-1_ (carbapenem resistance), *floR* (florfenicol/chloramphenicol resistance), *sul2* (sulfonamide resistance), *ble_MBL_* (bleomycin resistance), *msrE* (macrolide and phenicol resistance), *mphE* (macrolide resistance), *cmlB1* (phenicol resistance), and *aphA6* gene (aminoglycoside resistance) were found only in AMA_NO (Figure 2). The *aphA6* gene and the IS*Aba125* are located downstream of *bla_NDM-1_* in the Tn*AphA6* transposon. The *bla*_NDM-1_-Tn*AphA6* backbone was not found in any ST2 *A. baumannii* genomes, but it was present in 11 out of 3759 non-ST2 *A. baumannii* genomes. We analyzed the contig containing the *bla*_NDM-1_-Tn*AphA6* structure (contig 55) and compared it with genomic sequences in the GenBank database. A total of 101 sequences were identified with 98–100% nucleotide identity and 98–100% coverage. Most sequences belonged to plasmids (88 sequences), mainly identified in the genus *Acinetobacter* (49 sequences). However, other plasmid sequences were also identified in other genera, such as *Escherichia*, *Klebsiella*, *Citrobacter*, and *Providencia* spp. (Appendix A). Consequently, strain AMA_NO, as well as AMA166 are the first strains belonging to the ST2 clone to include the *bla*_NDM-1_-Tn*AphA6* array. The co-existence of *bla*_OXA-23_ and *bla*_NDM-1_ *A. baumannii* ST2 genomes occurs in a small percentage of these strains (63 out of 2938 ST2 genomes). Furthermore, only 109 *A. baumannii* ST2 genomes carry *bla*_NDM-1_. The *bla*_OXA-23_ gene has a greater representation among this group (521/2938 ST2 genomes). Both strains are resistant to ampicillin/sulbactam, cefepime, ceftazidime, ciprofloxacin, gentamicin, imipenem, and meropenem and are susceptible to minocycline, tigecycline, colistin, and amikacin (Table 2).

Virulence factors refer to traits (i.e., gene products) that allow a microorganism to establish itself on or within a host and enhance its potential to cause disease. Several literature reports of genome comparative analyses confirm the multifactorial and combinatorial nature of *A. baumannii* virulence [65,66,67,68,69,70,71,72]. A total of 46 genes associated with virulence factors were found in AMA_NO and AMA166 (Appendix A). *A. baumannii* produces a capsular polysaccharide (CPS) encoded by a gene cluster that is referred to as the K locus (KL). In contrast, the variable outer oligosaccharide of the lipopolysaccharides is encoded by the OC locus (OCL). CPS is an outer layer that is involved in protection against C3 deposition that occurs mainly in inhibiting macrophage phagocytosis. O antigen is responsible for the resistance of bacteria to complement mediated killing. Both components are essential to the blood passage of bacteria and the development of sepsis, but only CPS is involved in developing pulmonary infections [37,73,74,75]. The OCL5 and KL2 were found in AMA_NO and AMA166, respectively.

The human body has specific and non-specific defenses against infection. Early studies on the latter led to the discovery of the “hypoferremic response,” which consists of a reduction of free iron levels in blood and fluids in response to a bacterial invasion [76]. The consequence of the hypoferremic response is the iron starvation of invading bacteria. Later, it was found that iron is not the only essential element whose availability is reduced to interfere with bacterial growth. This understanding originated the concept of “nutritional immunity,” the group of non-specific defense strategies based on deprivation of the metal ions [77,78]. Bacteria must circumvent these nutritional limitations to establish in the body and cause disease. A common mechanism to overcome the lack of available iron is the biosynthesis of a siderophore that is secreted to the environment and competes with iron with high-affinity iron-binding proteins. Then, the iron-siderophore complexes are recognized by specific bacterial receptors to uptake iron [79,80].

*A. baumannii* iron uptake systems compete with high-affinity iron-binding host proteins to capture essential iron for the survival and progress of the infection. Antunes et al. identified six iron-uptake systems in *A. baumannii*. They are coded for by the acinetobactin locus, baumannoferrin locus, fimsbactin locus, Heme uptake cluster 2, Heme uptake cluster 3, and *feoABC* genes [81]. The acinetobactin locus, baumannoferrin locus, Heme cluster 2, and *feoABC* genes were found in AMA_NO and AMA166 (Appendix A). However, the acinetobactin and baumannoferrin loci were incomplete in both strains and are most probably nonfunctional. In the acinetobactin locus, the genes coding for *basE* (ACICU_02578) and a hypothetical protein (ACICU_02575) were missing. In the case of the baumannoferrin locus, there was a missing gene that encodes a siderophore synthetase (ACICU_01632). The four iron-uptake systems that were identified in *A. baumannii* AMA_NO and AMA166 were also found in 545 out of 2938 ST2 genomes (Appendix A). The complete acinetobactin and baumannoferrin loci were present in 2273 out of 2938 (77.36%) and 2839 out of 2938 ST2 genomes (96.63%), respectively. The complete sets of other systems were found in different percentages of ST2 genomes (Appendix A). While every ST2 genome contains at least one iron uptake system (Appendix A), there is heterogeneity in the combinations of systems that are present in each strain. A previous study that included 111 *A. baumannii* ST2 genomes found that they all harbor incomplete baumannoferrin and acinetobactin loci [82]. This homogeneity can be explained by the small ST2 genomes sample, which might not have been sufficient to represent the iron-uptake cluster variability of the ST2 clone accurately.

It must be noted that only the acinetobactin iron-uptake system has been linked to a high virulence phenotype [83]. The absence of acinetobactin in 665 ST2 strains could be the reason behind the low pathogenicity in some isolates of this clone (Appendix A).

### 3.3. Differential Expression of bla_NDM-1_ and bla_OXA-23_ in AMA_NO

Carbapenem-resistance in *Acinetobacter* spp. is usually due to the production of OXA-type carbapenemase and metallo-β-lactamases (MBLs), usually coded for by *bla*_OXA-23-like_, *bla*_OXA-58-like_, *bla*_OXA-51-like_, and the plasmid-mediated *bla*_NDM-1_ [84,85].

A recent study reported the co-existence of *bla*_OXA-23_ and *bla*_NDM-1_ in six isolates with ceftazidime and imipenem MIC values greater than 256 µg/mL and 32 µg/mL [86]. Quantitative RT-PCR (qRT-PCR) assays using total RNA extracted from *A. baumannii* AMA_NO cells that were cultured in LB showed two-fold higher *bla*_OXA-23_ mRNA levels compared to *bla*_NDM-1_ (Figure 4). Although other factors may impact the contribution of each gene to the resistance phenotype, the higher expression of *bla*_OXA-23_ may reflect a higher contribution to carbapenem resistance. It is also of interest that the *bla*_OXA-23_ gene was found in numerous *A. baumannii* clinical isolates but rarely in other Gram-negative species. Conversely, *bla*_NDM-1_ is a promiscuous gene, which could indicate a recent adaptive process. The genetic location of this gene can explain NDM global dissemination. Plasmids carrying *bla*_NDM_ have been described globally, with *Klebsiella* and *Escherichia* as the prevalent genus harboring them [87]. In *Klebsiella pneumoniae*, NDM has been reported in a wide variety of different STs worldwide, supporting its broad ability for dissemination [88].

The AMA_NO was isolated from a patient that was also infected with SARS-CoV-2. Bacterial/viral coinfections are not rare and are often synergistic, i.e., they produce enhanced symptomatic manifestations. Numerous pairs of bacteria/viruses that act synergically to cause or enhance infection have been described in the literature. A representative example is the *Streptococcus pneumoniae* and H1N1 influenza virus coinfection, which produce a lethal synergy [89]. The authors of this study observed that the bacterial infection induced a loss of lung repair responses. The fatal outcome was correlated with a loss of airway basal epithelial cells. Interestingly, the *S. pneumoniae* SirRH two-component system plays a role in the enhancement of bacterial survival inside lung cells that were previously infected with H1N1 [90]. Another mechanism that results in synergism is through the virus neuraminidase, which mediates the removal of sialic acid from the cell surface and facilitates the adhesion of invading bacteria [91]. Another significant observation is that some bacterial strains, when coinfecting with viruses, stimulated genetic recombination between two or more different viruses, acting as a driver of viral evolution. We hypothesize that the interactions between bacterial and viral elements could also result in bacterial acquisition of genes that increase virulence [92]. A future systematic study of bacterial isolates from SARS-CoV-2-infected patients may prove this hypothesis or not.

## 4. Conclusions

*A. baumannii* AMA_NO and AMA166 appear to belong to one clone. However, differences were identified in the pre- and post-pandemic strains, which could indicate adaptation to the environment found within the COVID patient. The SARS-CoV-2 viral infection could be a driver for evolutionary modifications. Expansion of genomic comparisons is needed to validate or disprove this hypothesis.

## Figures and Tables

**Figure 1 biology-12-00358-f001:**
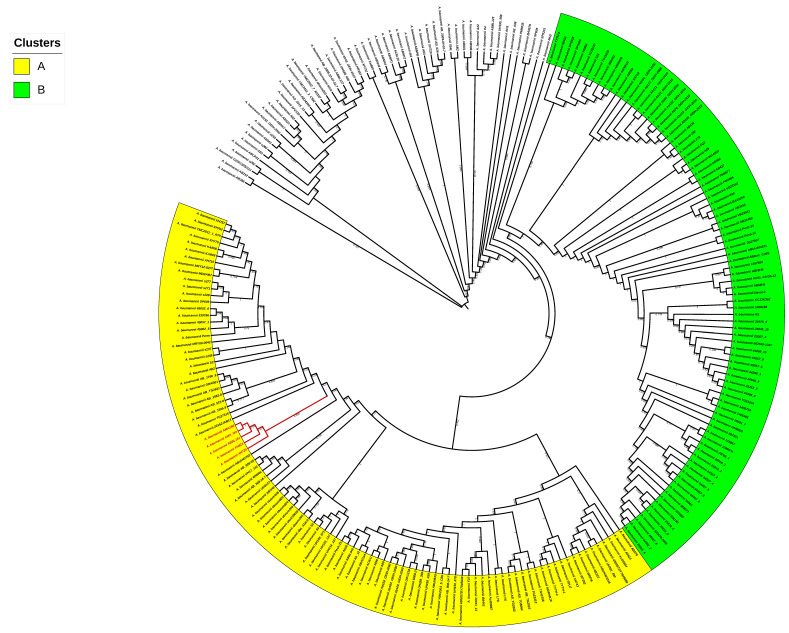
Core-genome phylogenetic analysis of AMA_NO, AMA166, and 239 representative genomes of ST2 *A. baumannii* clone. The figure displays the maximum likelihood phylogeny of 241 *A. baumannii* sequences. The bootstrap method was used as a supporting method (1000 iterations). The molecular substitution model was GTR. The tree representation was done by iTOL. Phylogenetic genomes related with AMA_NO and AMA166 is represented with the red branch. Yellow and green highlights represent A and B phylogenetic cluster, respectively. High resolution figures link: https://github.com/germant13/Traglia2023_AbST2_SARS (accessed on 9 February 2023).

**Figure 2 biology-12-00358-f002:**
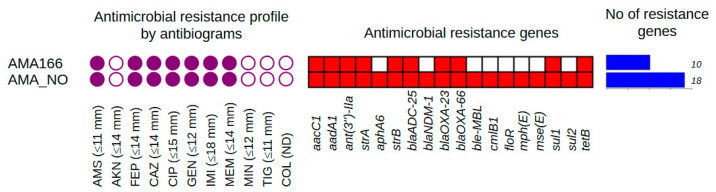
Comparison of antimicrobial susceptibility profile and antimicrobial resistance genes among AMA_NO and AMA166 strains. The figure is deposited in GitHub Repository (https://github.com/germant13/Traglia2023_AbST2_SARS.git (accessed on 9 February 2023).

**Figure 3 biology-12-00358-f003:**
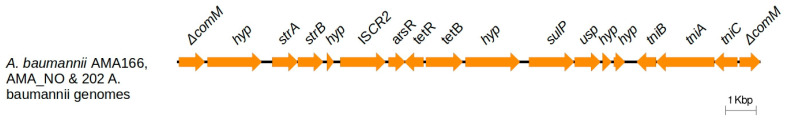
Genetic structure of Tn*AbaR* genomic island present in *A. baumannii* AMA_NO and AMA166. An identical genomic island was present in both strains. Figure 3 is deposited in GitHub Repository (https://github.com/germant13/Traglia2023_AbST2_SARS.git (accessed on 9 February 2023).

**Figure 4 biology-12-00358-f004:**
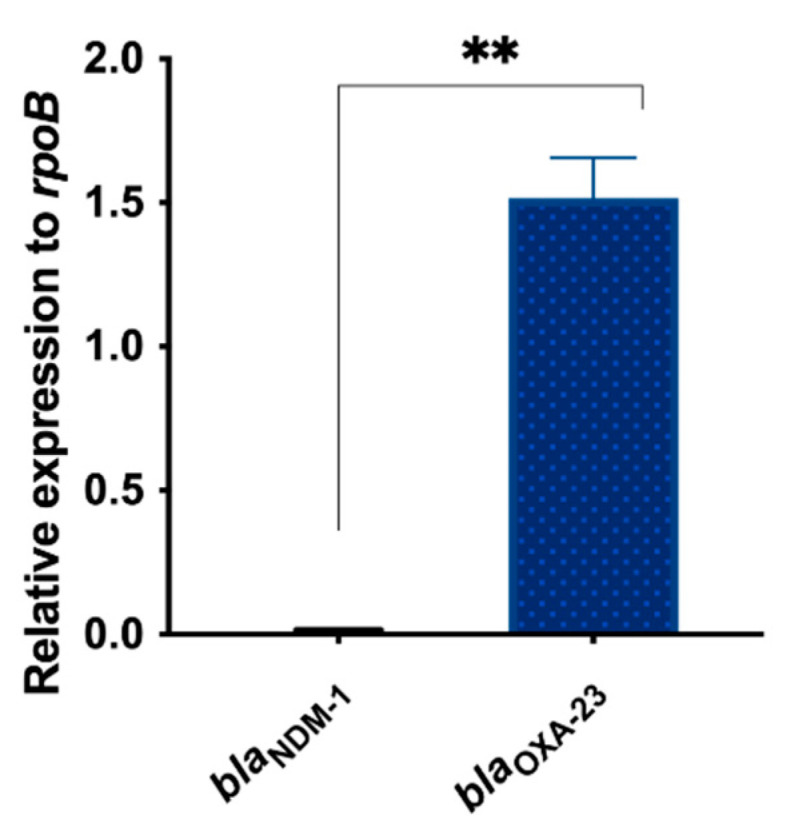
Expression analysis by qRT-PCR of genes coding for β-lactamases (*bla*_OXA-23_ and *bla*_NDM-1_) in the AMA_NO strain. Fold changes were calculated using ΔCt analysis. At least three independent samples were used, and four technical replicates were performed from each sample. Student’s *t*-test analysis was performed using GraphPad Prism (GraphPad software, San Diego, CA, USA). **: A *p*-value < 0.01 was considered statistically significant. Data are presented as the mean ± SD.

**Table 1 biology-12-00358-t001:** Genome features of *A. baumannii* AMA_NO and AMA166.

Strains	Isolates Date	Genome Size (bp)	GC% Content	N50	N Contig	Coverage Depth	MLST Profile/Clonal Complex	tRNA	ncRNA	NCBI Accession Number
AMA166	2016	3.823.583	39	220435	38	108X	ST2/CC2	60	38	JANKJZ000000000
AMA_NO	2021	4.078.010	39	125130	79	80X	ST2/CC2	61	41	JANKKA000000000

**Table 2 biology-12-00358-t002:** Antibiotic susceptibility in *A. baumannii* AMA 166 and AMA_NO.

	Diameters of Inhibition Zones (mm)/Minimum Inhibitory Concentrations (MICs)	
	AMA 166	AMA_NO
Ampicillin/sulbactam (AMS)	6 (R)	6 (R)
Amikacin (AKN)	24 (S)	19 (S)
Cefepime (FEP)	6 (R)	6 (R)
Ceftazidime (CAZ)	19 (R)	6 (R)
Ciprofloxacin (CIP)	6 (R)	6 (R)
Gentamicin (GEN)	8 (R)	7 (R)
Imipenem (IMI)	6 (R)	6 (R)
Meropenem (MEM)	6 (R)	6 (R)
Minocycline (MIN)	20 (S)	22 (S)
Tigecycline (TIG)	20/0.5 mg/L (S)	20/1 mg/L (S)
Colistin (COL)	14/1 mg/L (S)	14/1 mg/L (S)

S: susceptible, R: resistant diameters of inhibition zones of antibiogram plates performed according to CLSI. The experiments were repeated at least three times for each strain. The results were interpreted with CLSI guidelines, except colistin (EUCAST) and tigecycline (FDA). The assays were performed by duplicates.

## Data Availability

The Whole Genome Shotgun project has been deposited at GenBank with accession numbers JANKJZ000000000 and JANKKA000000000 for AMA166 and AMA_NO, respectively.

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
