# Peer review of "Genomic Comparative Analysis of Two Multi-Drug Resistance (MDR) Acinetobacter baumannii Clinical Strains Assigned to International Clonal Lineage II Recovered Pre- and Post-COVID-19 Pandemic"

_biology, 2023, doi:10.3390/biology12030358_

Round 1

Reviewer 1 Report

This article deals with a comparative genomic analysis of A baumanii, it has a great novelty in terms of sars cov2 pandemic, however it lacks of a critical evaluation of their results in terms of biological and ecological relationships of their strains, authors have a great topic, but their manuscript must be improved to be publishable at biology journal.

My detailed comments are in the attached file

Author Response

We appreciated the reviewer’s suggestions. We have considered them and we included our responses to their questions/comments in the uploaded document.   We hope that you now find the manuscript suitable for publication in Biology, thanks

Reviewer 2 Report

In this manuscript, the authors present genomic analysis and comparison of two Acinetobacter baumannii strains isolated pre and post the Covid19. And based on the comparison results, they hypothesized that the gene differences could result from adaptive evolution to the human body infected with SARS-CoV-2. I suggest giving a deep and detailed discussion about whether viral infection could be the evolutionary factor resulting in the genetic variation of two isolates., such as the possible interaction of the co-infection. As there could be many environmental factors involved in the pathogenic evolution within this time period.

Some other comments and questions:

-Acronyms/Abbreviations/Initialisms should be defined the first time they appear in each of three sections: the abstract; the main text; the first figure or table. 

-Line 97 and 213 Antimicrobial resistance gene and antibiotic resistance gene were both abbreviated as ARG. To minimize confusion, please keep the definition consistent.

-Line 235  “aphA6”, italic

-Line 252 LPS, no need to abbreviate, as this is the first and last time mentioned in the manuscript

-Line 264 “et al”, italic 

-Figure 1 The resolution of this figure needs to be improved. None of the text is readable. Add a co-linearity analysis between two pre and post COVID-19 isolates.

-Figure 2 The spacing within words needs to be adjusted

-Table 2 Is there parent strain used as control for the antibiotic susceptibility? What do the values of diameters of inhibition zones stand for? Mean or median of multiple experiments? This needs to be clarified. Check spelling, “Minimum Jnhibitory”.

-Figure 4 What does “**” mean? Does it mean p<0.01?

Author Response

(The authors gave the same response as above.)

Reviewer 3 Report

Manuscript ID: biology-2156259 is an interesting and well-written study on the genomic comparative analysis of two multi-drug resistance Acinetobacter baumannii clinical strains assigned to international clonal lineage II recovered pre- and post-COVID-19 pandemic. The study adds novel information to the field and is of interest for the readers of the journal. I am listing below queries which Authors might address to increase the scientific relevance of the study.

 1. The title of the manuscript should be better changed into “Genomic comparative analysis of two multi-drug resistance Acinetobacter baumannii clinical strains assigned to international clonal lineage II recovered pre- and post-COVID-19 pandemic”.

2. Two MLST schemes are available for Acinetobacter baumannii, the Pasteur scheme and the Oxford scheme. Of these, the Oxford scheme is more discriminant than the Pasteur MLST scheme and is able to identify additional genotypes and to differentiate isolates belonging to international clone II into at least three distinct clades/clonal lineages. However, the Oxford scheme is affected by recombination and artifactual issues, such as the presence of gdhB paralog. Nevertheless, Authors should genotype the two A. baumannii clinical isolates described in the study and the collection of genomes assigned to Pasteur ST2 included in the study using the Oxford MLST scheme also. Interestingly, MLST typing of the two genomes from the clinical isolates pre and post-COVID-19 pandemic are assigned to Oxford ST208, which is one of the three epidemic clades which Oxford MLST scheme is able to differentiate among Pasteur ST2 allelic profile. Authors can refer for additional information to Gaiarsa et al. Comparative Analysis of the Two Acinetobacter baumannii Multilocus Sequence Typing (MLST) Schemes. Front Microbiol. 2019;10:930. doi: 10.3389/fmicb.2019.00930

 3. blaNDM-1 in A. baumannii AMA_NO. It should be interesting to demonstrate if blaNDM-1 acquisition in  A. baumannii AMA_NO isolate occurred through conjugation. This can be assessed by performing classical mating experiment using A. baumannii AMA_NO isolate as donor and reference A. baumannii ATCC19606 or E. coli as recipient strains. Authors should consider also to demonstrate it in silico by analyzing genomic regions flanking blaNDM-1 gene into JANKKA000000000 whole genome sequences and its localization into a plasmid. In partial support of this, BLASTn analysis performed by this reviewer maps blaNDM-1 to contig 55 flanked by bleomycin resistance gene and IS sequences. Also, the 4411 nucleotide sequences of contig 55 are 100% identical to those found in conjugative plasmid pAB17, NCBI Ref Seq: NZ_MT002974.1, from A. baumannii strain AB17. Authors should comnsider to use Bandage software (Wick et al. 2015; Bioinformatics 31, 3350–3352) to identify and connect plasmid contigs into JANKKA000000000 WGS.

 4. To study the genetic distance as a function of the collection time between JANKJZ000000000 and JANKKA000000000 genomes recovered pre- and post-COVID-19 pandemic, Authors should consider to perform Beast Timing Analysis of A. baumannii strains assigned to Oxford ST208 for which the collection date is listed under the biosample data on NCBI. A regression analysis implementing root-to-tip genetic distance as a function of the sample collection year can be conducted on core-SNP matrix and a measure of clocklike behavior can be assessed.

5. "CLSI M100-ED32:2022 Performance Standards for Antimicrobial Susceptibility Testing, 32nd Edition”. Is this the correct quotation for reference 46 ?

Author Response

(The authors gave the same response as above.)

Round 2

Reviewer 1 Report

The authors have addressed all my concerns and suggestions; just they need to double-check for minor spell mistakes

Author Response

Reviewer #1 (Comments for the Author):

- The authors have addressed all my concerns and suggestions; just they need to double-check for minor spell mistakes

Author’s response:

As advised, we did a double-check for minor spell mistakes and fixed them. We appreciate that Reviewer’s comments and decision.

Reviewer 2 Report

Further comments:

- Line 380 "S. pneumoniae" italic

- Uniform the reference format. Some of the reference titles are capitalized in every word (for example, #8 Peleg AY, Hooper DC (2010) Hospital-Acquired Infections Due to Gram-Negative Bacteria. New England Journal of Medicine 362: 1804–1813.), while others are capitalized in the first word ( for example, #9 Liao YT, Kuo SC, Lee YT, et al. (2014) Sheltering effect and indirect pathogenesis of carbapenem-resistant Acinetobacter baumannii in polymicrobial infection. Antimicrob Agents Chemother 58: 3983–3990.).

Author Response

Reviewer #2 (Comments for the Author):

- Line 380 "S. pneumoniae" italic

Author’s response:

Fixed.

- Uniform the reference format. Some of the reference titles are capitalized in every word (for example, #8 Peleg AY, Hooper DC (2010) Hospital-Acquired Infections Due to Gram-Negative Bacteria. New England Journal of Medicine 362: 1804–1813.), while others are capitalized in the first word (or example, #9 Liao YT, Kuo SC, Lee YT, et al. (2014) Sheltering effect and indirect pathogenesis of carbapenem-resistant Acinetobacter baumannii in polymicrobial infection. Antimicrob Agents Chemother 58: 3983–3990.).

Author’s response:

We uniformed the format. Thanks for noticing it.

Reviewer 3 Report

Authors correctly addressed all issues raised by reviewers and modified the manuscript accordingly. As minor comment, I ask Authors to correct quotation of reference nr. 59: Gaiarsa et al. (2019). Front. Microbiol. 10:930. I suggest also minor English Editing. 

Author Response

Reviewer #3 (Comments for the Author):

Authors correctly addressed all issues raised by reviewers and modified the manuscript accordingly. As minor comment, I ask Authors to correct quotation of reference nr. 59: Gaiarsa et al. (2019). Front. Microbiol. 10:930. I suggest also minor English Editing.

Author’s response: We appreciated the reviewer’s feedback. We change the quotation of the reference and performed English editing.